# Considering patient clinical history impacts performance of machine learning models in predicting course of multiple sclerosis

**Ruggiero Seccia**[1], **Daniele Gammelli**[1¤], **Fabio Dominici**[1], **Silvia Romano**[2], **Anna Chiara Landi**[2], **Marco Salvetti**[2,3], **Andrea Tacchella**[4], **Andrea Zaccaria**[4], **Andrea Crisanti**[5], **Francesca Grassi**[6☯*], **Laura Palagi**[1☯*]

**1** Dept. of Computer, Control and Management Engineering Antonio Ruberti, Sapienza University of Rome, Rome, Italy, **2** Dept. of Neurosciences, Mental Health and Sensory Organs, Sapienza University of Rome, Rome, Italy, **3** IRCCS Istituto Neurologico Mediterraneo Neuromed, Pozzilli, Italy, **4** Dept. of Physics, Istituto dei Sistemi Complessi (ISC)-CNR, UOS Sapienza, Rome, Italy, **5** Dept. of Physics, Sapienza University of Rome, Rome, Italy, **6** Dept. of Physiology and Pharmacology, Sapienza University of Rome, Rome, Italy

☯ These authors contributed equally to this work.
¤ Current address: Technical University of Denmark, Kongens-Lyngby, Denmark
* francesca.grassi@uniroma1.it (FG); palagi@diag.uniroma1.it (LP)

**Data Availability Statement:** The data underlying the results presented in the study is available on Figshare (https://doi.org/10.6084/m9.figshare.

## Abstract

Multiple Sclerosis (MS) progresses at an unpredictable rate, but predictions on the disease course in each patient would be extremely useful to tailor therapy to the individual needs. We explore different machine learning (ML) approaches to predict whether a patient will shift from the initial Relapsing-Remitting (RR) to the Secondary Progressive (SP) form of the disease, using only "real world" data available in clinical routine. The clinical records of 1624 outpatients (207 in the SP phase) attending the MS service of Sant'Andrea hospital, Rome, Italy, were used. Predictions at 180, 360 or 720 days from the last visit were obtained considering either the data of the last available visit (Visit-Oriented setting), comparing four classical ML methods (Random Forest, Support Vector Machine, K-Nearest Neighbours and AdaBoost) or the whole clinical history of each patient (History-Oriented setting), using a Recurrent Neural Network model, specifically designed for historical data. Missing values were handled by removing either all clinical records presenting at least one missing parameter (Feature-saving approach) or the 3 clinical parameters which contained missing values (Record-saving approach). The performances of the classifiers were rated using common indicators, such as Recall (or Sensitivity) and Precision (or Positive predictive value). In the visit-oriented setting, the Record-saving approach yielded Recall values from 70% to 100%, but low Precision (5% to 10%), which however increased to 50% when considering only predictions for which the model returned a probability above a given "confidence threshold". For the History-oriented setting, both indicators increased as prediction time lengthened, reaching values of 67% (Recall) and 42% (Precision) at 720 days. We show how "real world" data can be effectively used to forecast the evolution of MS, leading to high Recall values and propose innovative approaches to improve Precision towards clinically useful values.

11902854). Please note that we have removed the labels to most features, to protect privacy of study participants to the maximum possible extent. This action will not preclude duplication of our results.

**Funding:** LP acknowledges partial support from the project 'MIME-BCI: Mindfulness Meditation training supported by Brain-Computer Interfaces" (2016) (No PI11161550696379A) from Sapienza, University of Rome (www.uniroma1.it). The funders had no role in study design, data collection and analysis, decision to publish, or preparation of the manuscript. LP and RS acknowledge financial support from the Sapienza University of Rome (www.uniroma1.it) within the project "Network medicine-based machine learning and graph theory algorithms for precision oncology" (2019). The funders had no role in study design, data collection and analysis, decision to publish, or preparation of the manuscript AT and AZ acknowledge financial support from the Italian Ministry of University and Research (www.miur.gov.it) within the project CRISIS LAB PNR 2011-2013. The funders had no role in study design, data collection and analysis, decision to publish, or preparation of the manuscript.

**Competing interests:** The authors have declared that no competing interests exist.

## Introduction

Multiple sclerosis (MS) is an inflammatory disease of the central nervous system that typically starts with a relapsing-remitting (RR) phase that gradually turns into a secondary progressive (SP) form, in which disability accumulates. However, the natural course of the disease is extremely variable, ranging from extremely mild to very aggressive forms. In particular, the duration of the RR phase is quite variable, and relapses occur randomly [1]. Given the clinical heterogeneity of multiple sclerosis, reliable prognostic predictors would be of great importance. Several prognostic factors of disability have been described, and some studies have proposed risk scores calculated from demographic and clinical variables collected at disease onset [2, 3]. However, the prediction of the course of MS on the basis of clinical and other supportive data is challenging, and no validated prediction model for the clinical course is currently available. To date, several clinical and demographic features associated with long-term disease course have been proposed. Older age at onset [4, 5] and male sex [4, 6] have been associated with an increased risk of disability progression in the long-term. Family history of MS seems to be a predictor of a shorter time to conversion to SPMS [7]. Environmental and modifiable factors, such as smoke and high body mass index, contribute to impairments in walking speed, overall disability and depression [8]. Vitamin D deficiency has been associated with worse outcome as well [9]. Poor prognosis also correlates with high annualized relapse rate, particularly on-treatment, short interval between disease onset and first relapse [10, 11], incomplete recovery from the first relapse or polysymptomatic onset [4, 6]. Some studies have highlighted the role of the localization of the first relapse. Motor onset and early cerebellar involvement have been associated with faster increase in disability, while sensory onset and optic neuritis have been described as favourable prognostic factor [4, 12]. Radiologic biomarkers of prognosis include a high number of T2 lesions at baseline MRI, whole-brain and grey matter atrophy observed in the earliest stages, presence of cerebellar and brainstem lesions and increased T2 lesion number or lesion volume within the first 2 years [7, 13, 14]. Some studies have suggested that the presence of oligoclonal bands in the cerebrospinal fluid at the time of diagnosis predicts a worse prognosis with high disability impairment [15, 16].

On the other hand, several preventive disease-modifying therapies (DMTs) are available nowadays, so that, in principle, it is possible to tailor treatments to the specific needs of each patient. Considering that all therapies are preventive, their costs and their safety profile [17], it would be extremely useful to have prognoses as exact as possible, to avoid under-treatment of patients with aggressive forms of disease or over-treatment of patients with mild forms.

For support to patient counseling, prognosis, and therapy, attention has increasingly been turned to artificial intelligence, exploiting the ability of Machine learning (ML) approaches to extract complex relations among existing data without requiring *a priori* models linking input and output variables [18].

Different problems have been addressed, such as the classification of disease phase at the time of analysis [19–21] or evaluation of the probability of transition from Clinically Isolated Syndrome (CIS) to definite multiple sclerosis within 1 to 3 years [22–24]. Others have attempted to derive predictions on the course of individual patients or have investigated the variables that best predict disease evolution in time [25–28].

Several of these works relied on imaging data, a factor likely to limit the diffuse application of the proposed method, as algorithms might be hardly able to understand images obtained using different devices [29]. Moreover, none of the proposed prognostic methods achieved Specificity and Sensitivity of "clinical grade".

To date, the use of learning machines remains outside the clinical practice, for the above reasons and possibly also because of the limited confidence of physicians with the technology and the absence of collaboration between computer people and clinicians [29–31].

To address some of these possible limiting factors it is important to produce tools that are perceived as user-friendly by clinicians. One important step is to use data that are readily available in normal clinical practice ("real-world" data), so that doctors can autonomously use the machines once they are built, without breaking their (already hectic) working routine. Thus, in this paper we explore the possibility of predicting whether a patient will pass from RR to SP phase in a given time window, using a real-world dataset, built in close collaboration between computer experts and neurologists. The database contains the results of neurological and imaging exams routinely collected during the periodic visits. The use of routinely available clinical data might help to spread the usage of ML among physicians, as the proposed method can be replicated in any center where a database is available. Moreover, the use of clinical registers allows the exploitation of long term data, which is critical for a chronic, long-lasting disease such as MS [32].

Raw data are the collection of many clinical records (samples), each containing information on a number of clinical parameters (features) and on the MS phase of the patient, either RR or SP at the time of the visit. Since any real-world raw dataset is plagued by incomplete records, data pre-processing represents a crucial step in such an analysis. Keeping in mind the clinical relevance of the data, we processed raw data in two different ways, producing datasets that contain the maximum number of either records (Record-Saving dataset, RS) or features (Feature-Saving dataset, FS). Either way, this unavoidable cleaning reduces the amount of available data, already limited at the beginning, increasing the challenge of using ML methods on a collection of "small data", rather than on "big data".

To predict whether a patient will pass from the RR to the SP phase within a certain time-window (180, 360 or 720 days), we used two different approaches. The first is a "spot" approach: predictions are based on the results of a single visit, so that a sample in the dataset corresponds to a visit of a patient. The second, instead, considers the clinical history of the patient up to the last available visit, namely each sample of the training dataset consists in a sequence of consecutive visits. Hereafter, we will refer to the former approach as the *Visit-Oriented* (VO) setting, while to the latter as *History-Oriented* (HO) setting.

Finally, for both the VO and the HO approach we used the RS and FS datasets to train several machine learning algorithms to identify the most reliable and promising one(s), to understand the impact of different data processing strategies on the models performance and to define the importance of the amount and type of available data for training the machine learning models. To favour the spreading of our approach, we chose broadly available ML procedures (Random Forest, Support Vector Machine, K-Nearest Neighbours, AdaBoost, Long short-term memory neural networks), that can be easily implemented everywhere.

The time-windows chosen for the predictions are 180, 360 and 720 days. Possibly, the predictions over two years (720 days) are the most relevant from a clinical point of view. Nevertheless performance on shorter times can improve the model analysis, for instance by giving information on the the more relevant features and/or on the reliability of the models.

To the best of authors' knowledge, this work represents the first attempt to analyse different approaches to prepare sets of routine clinical data, learning strategies and machine learning models to forecast the evolution of MS course.

The analysis of the predictive *confidence* of the classification models appears a promising tool to boost the performance of data-driven models. Future work will be able to exploit the power of the HO setting to extend the analysis of larger datasets over longer time windows.

## Materials and methods

### Dataset description

Data has been provided by the Multiple Sclerosis centre of Sant'Andrea University hospital in Rome. The use of the database for research purposes was authorized by the Ethical committee of Sapienza University (Authorization n. 42542016, November 2, 2016). All patients provided written informed consent to have anonymous data from their medical records used in scientific research. At each visit, patients underwent neurological examination. The clinical status, laboratory and image data, when available, and the phase of the disease (RR or SP, as scored by the attending neurologist) were entered in a database. If considered necessary, neurologists were able to modify the recorded disease staging during subsequent visits.

The raw dataset is composed of 18 574 clinical records from visits performed between 1978 and 2018 on 1 624 patients, of whom 207 (12.7%) in SP phase, now followed at the MS Center of Sant'Andrea University hospital, designated and certified as a MS center of excellence. The percentage of SP patients is just below the lower end of international reference values (13.5-32%) obtained using nationwide cohorts [33–37]. A likely explanation for the lower transition rate is that, as a part of the peculiar routine inherent to the excellence center, patients are treated early with second line drugs, and this approach may delay the SP transition. Furthermore, the median onset times of SPMS is 23-34 years from disease onset [33, 34], while average disease duration in the cohort under study is 19 years, contributing to reduce the percentage of patients with SPMS.

Each clinical record (sample) contains up to 200 fields (features) describing the status of the patient. The number of visits for each patient varies between 1 and 56 (Table 1). The dataset shared for analysis was anonymized, by removing all features potentially revealing patients' identity (such as name, tax code, address).

For the purpose of the construction of our prediction models, to each sample $i$ we added three fields with binary outcomes (the output labels), denoted as $y_i^{180}$, $y_i^{360}$ and $y_i^{720}$, which tell whether the patient passed to the SP phase of the disease after 180, 360 or 720 days from the visit $i$, respectively. Formally, we define the outcomes as follows:

$$y_i^k = \begin{cases} 1, & \text{patient passes to the SP phase within } k \text{ days after the visit } i \\ 0, & \text{otherwise} \end{cases}$$

### Data preprocessing

Since we are interested in predicting the *transition* of a patient from the RR to the SP phase, rather than simply *assigning* patients to the RR or SP phase, we removed from the dataset all the visits occurred after the transition of the patient to the SP phase. As a consequence we obtain three datasets for the prediction of the status after 180, 360, 720 days, which may have different number of samples.

Preprocessing was necessary to account for missing *(not available, NA)* data and to encode descriptive variables in terms amenable to computer analysis. As a preliminary preprocessing phase, we deleted those features characterized by many categorical values (e.g. city of birth), which increase the dimension of the dataset without providing really useful information and those features only occasionally present.

Through these procedures, the dimensionality of the dataset was reduced from more than 200 features to only 21 that can be divided into four main categories: Demographic, Clinical data and relapses, Magnetic Resonance Imaging (MRI) and Liquor analysis, Therapeutic treatments.

**Table 1. Statistics on the raw dataset.**

|  | Total | Mean | SD | Min | Max |
|---|---|---|---|---|---|
| Total # of patients | 1624 | - | - | - | - |
| Male | 490 | - | - | - | - |
| Female | 1134 | - | - | - | - |
| SP patients | 207 | - | - | - | - |
| Years of observations | - | 6 | 5 | 0 | 32 |
| Age at onset | - | 29 | 9 | 8 | 63 |
| Total # of visits | 18574 | - | - | - | - |
| # of visits per patient | - | 11 | 8 | 1 | 56 |

#: Number; SD: Standard deviation

As regard the therapeutic treatments, being there hundreds of different drugs potentially assumed by patients, this information was organized into 6 different clusters: *Relapse treatment* drugs; *MS symptomatic treatment* drugs; *First line Disease Modifying Therapies* (DMT), which include Interferons, Glatiramer Acetate, Teriflunomide and Dimethyl fumarate; *Second line DMTs*, such as Fingolimod, Natalizumab, Alemtuzumab, and Rituximab; *Immunosuppressants*, including Azathyoprine, Mitoxantrone, Cyclophosphamide and Methotrexate; *Other treatments* (related to concomitant pathologies). The main difference between second line DMTs and immunosuppressant therapy is the selectivity of action. The latter are classical cytotoxic immunosuppressants that act by inhibiting DNA synthesis [38, 39], while second line drugs act by suppressing or altering the immune system with more specific mechanisms [40].

The final features in our dataset are defined in Table 2.

Despite the cleaning procedure described above, more than 14 000 samples (75% of the total) included one or more missing value, all of them corresponding to the features: Status T1 (6.632 empty fields, 35% of total), Status T2 (3.569, 19%) and Oligoclonal Banding (10.887, 57%). Such a situation is reasonable from a clinical point of view, as not all visits are accompanied by Magnetic resonance imaging (features Status T1 and Status T2), nor all patients undergo lumbar puncture (feature Oligoclonal banding).

To cope with these cases, elimination of NA fields was the only available possibility, so two distinct NA-elimination strategies were implemented: a *Feature-saving* (FS) strategy, consisting in eliminating all the records in which at least one NA value was present in the attempt to exploit all the clinical information collected in each visit; a *Record-saving* (RS) strategy, consisting in eliminating the three features where the NA values occurred, preserving all the records and hence the full amount of visits. These two procedures yielded to two different datasets, with 21 or 18 features, respectively. The number of records included depends on the timespan considered for the prediction of the transition to SP phase (180, 360 or 720 days). Table 3 summarises the number of entries in each of of the six datasets so obtained. Some of these features, such as age or sex, have been already suggested to be predictors of the rate of disability progression (see Introduction). Other putative prognostic factors, such as vitamin D deficiency or body mass index, were not considered due to lack of recorded data.

We performed a further statistical analysis based on the Pearson matrix to check for potential correlations among features and/or among each feature and the output. We used this analysis to:

- identify potentially redundant input features, discarding those that could be deduced by other features or their combination.

**Table 2. Features used for training the machine learning models.**

| Type | Variables |
|------|-----------|
| Demographic | Age at onset |
| | Gender |
| | Age at Visit |
| Clinical Features | EDSS |
| | # Relapses from last visit |
| | Pregnancy |
| | Relapses frequency |
| | Time from last relapse |
| MRI and liquor | Status T1 |
| | Status T2 |
| | Spinal Cord |
| | Supratentorial |
| | Optic Pathway |
| | Brainstem-Cerebellum |
| | Oligoclonal Banding |
| Therapeutic treatments (drugs) | Relapse treatment drugs |
| | First line DMT |
| | Immunosuppressant |
| | MS symptomatic treatment drugs |
| | Second line DMT |
| | Other drugs |

EDSS: expanded disability status scale; Status T1/T2: Presence of gadolinium-enhanced lesions in T1/T2; Spinal Cord, Supratentorial, Optic Pathway, Brainstem-Cerebellum: Presence of lesions in the corresponding regions; Oligoclonal banding: detection of of oligoclonal bands in liquor.

- analyse the influence of the features with respect to the output labels

As a matter of example we report in Fig 1 the Pearson Matrix for the Dataset FS at 180 days describing the relation among input features and the transition to the SP phase. This analysis highlighted a very mild correlation between the output label and the features that in most cases is equal to 0 (gray squares). The features that show the most relevant correlation, which remains very mild (absolute value of the Pearson coefficient below 0.2), with the output are: Time from last relapse, EDSS, Relapses Frequencies, Age at onset, Oligoclonal Banding and Status T2.

**Table 3. Composition of the Feature-saving and Record-saving datasets.**

| Days | Strategy | Features | Records | Patients | SP patients | % SP Records |
|------|----------|----------|---------|----------|-------------|--------------|
| 180 | FS | 21 | 4330 | 506 | 36 | 0.8 |
| | RS | 18 | 14923 | 1515 | 207 | 1.3% |
| 360 | FS | 21 | 4202 | 495 | 37 | 0.8% |
| | RS | 18 | 14238 | 1449 | 207 | 1.4% |
| 720 | FS | 21 | 3923 | 468 | 37 | 0.9% |
| | RS | 18 | 13178 | 1375 | 207 | 1.5% |

The 6 dataset were obtained through the FS or RS *NA-elimination* strategies for each of the timespan considered in the classification task.

Given the slow course of MS and the fact that we maintained only one record per patient after the transition to the SP phase, most records pertain to patients in the RR phase. Thus, as shown in Table 3, the classification problem is highly imbalanced, as the number of records relating to patients after transition to the SP phase is never higher than 1.5% of the total number of records. This issue makes the classification task significantly more complex and has been tackled during the learning process by means of specific techniques that will be outlined in the following section.

### Data analysis

The analysis aimed to learn a function that partitions the data into two groups which in our case were the patient either in the RR or in the SP phase. The binary classification problem was addressed in two different settings, a Visit-Oriented setting, and an History-Oriented setting.

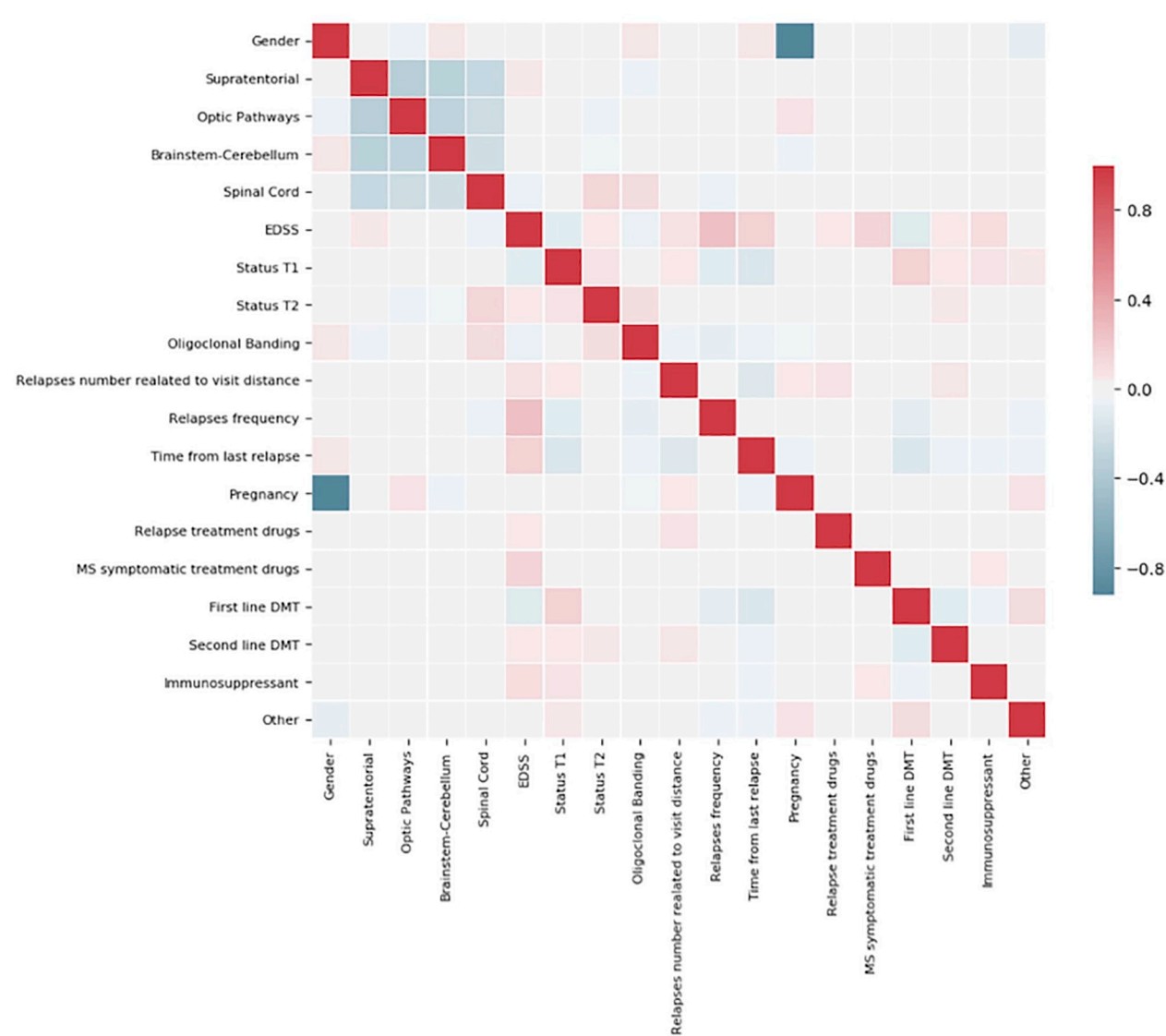

**Fig 1. Pearson Matrix describing the relation among input features and the transition to the SP phase within 180 days.**

The difference relies on whether the single visit or the clinical history of the patient is considered in predicting the outcome $y_i^k$.

**The Visit-Oriented setting.**   In the Visit-Oriented (VO) setting predictions were done based on the information contained in individual clinical records, that is, gathered during a single visit. More formally, every record is considered as an isolated point in the *feature space*. The dataset for the three time windows $k = 180, 360, 720$ in this case are made up of pairs $(x_i, y_i^k)$ with $i = 1, \ldots,$ #visits$^k$ and $x_i \in R^n$ where $n = 21, 18$ respectively for the FS or RS strategy and $y_i^k \in \{0, 1\}$.

We trained different machine learning models and we compared the performance achieved using standard Key Performance Indicators (KPIs) as specified below. The ML models which have been considered are: Random Forest (RF) [41], Nonlinear Support Vector Machines (SVM) [42], K-Nearest Neighbours (KNN) [43], AdaBoost (AB) [44]. Computations were performed using the open source package SKlearn, on Python 3.7. For lack of space, these methods are not described here. A discussion about these machine learning algorithms can be found in [45] and references therein.

Being the data highly unbalanced, a standard train-validation-test split of the datasets was not reliable because it would yield a very low estimate of the number of patients transitioning to the SP phase. Therefore, a problem-specific cross-validation procedure was implemented in order to assess the overall performance of the models and give a reliable estimate of the generalization capabilities of the models to out-of-sample data. Moreover, since particular features might allow some models to uniquely identify patients (leading therefore to an algorithm capable of *recognizing* the patient rather than extrapolating meaningful regularities in the data), it was necessary to avoid that records belonging to the same patient were present in both the training and the test set. For this reason, a **Leave One Group Out** (LOGO) cross-validation procedure was implemented, where the test set was iteratively defined by excluding all the visits of a single patient from the training set. Given this framework, every iteration of the cross-validation procedure would see the models testing their predictions on new unseen patients.

To tackle the unbalanced structure of the data, each model was trained by means of the Bootstrap Aggregating (**Bagging**) procedure [46]. In Bagging, $B$ different balanced subsets are built through a subsampling of the initial dataset, then each subset is used to train a model. The $B$ models so obtained are then tested on the patient left out by means of the LOGO cross-validation and the final prediction is built as an average of the predictions returned by the $B$ models.

Overall, the training procedure can be formalized as follows:

- Take out all the visits belonging to one patient (this will be the test fold)

- Extract $B$ balanced sub-dataset and train $B$ models, one for each sub-datasets (Bagging procedure)

- Evaluate the performance of the trained models as an average of the predictions on the visits belonging to the "validation patient"

- Repeat this procedure for all the patients in the dataset

Finally, in order to determine the best set of hyperparameters for each model, a **Grid Search** was carried out, where the leading performance indicator for choosing the best set of hyperparameters was the Recall (see below for details on performance indicators).

**History-Oriented setting.**   In the History-Oriented (HO) setting, each record is considered in an overall clinical history of a patient. Algorithmically, each patient is treated as a time series representing a specific course of MS and each record is used together with all the

previous clinical records of the same patient to determine the prediction of the model. This approach was implemented by applying a Long Short-Term Memory (LSTM) [47–49] neural network with 10 cells followed by a Feedforward Neural Network with a sigmoidal (logistic) function as activation function in output. Computations were performed using the open source package Keras, on Python 3.7. In order to improve the generalization properties of the model, we introduced a Dropout probability equal to 0.2.

Because of the computational burden of these particular models, the LOGO procedure was not feasible in terms of computing time and therefore a standard train-test split was performed using the former set for training the model and the latter to check its generalization performance. Note that training and test sets were built in such a way that the proportion of patients passing to the SP phase was kept unchanged. For the same computational constraint reasons, hyperparameter optimization was not carried out like in the Visit- Oriented approach but the number of neurons was chosen through a trial and error procedure. The Bagging procedure, instead, given its evident relevance in mitigating the unbalanced structure of the classification problem, was implemented as for the Visit-Oriented scenario.

**Metrics for comparison.** In order to determine the quality of the machine learning models, different indicators were considered. The need for taking into consideration different metrics of evaluation is due to the fact that datasets are unbalanced and type I errors (False positive) and type II errors (False Negative) have a different meaning [50]. In this section, we introduce the main metrics taken into account and explain their meaning and importance in the multiple sclerosis-specific setting. Although other metrics could be taken into consideration, in this work we focused on the more relevant from a prognostic point of view.

*Confusion Matrix* Predictions and real occurrences are reported in an aggregated manner through a matrix, where the rows represent the true labels and the columns the model predictions, as reported below.

| | | Predicted | |
|---|---|---|---|
| | | Non-transiting | Transiting |
| True | Non-transiting | TN | FP |
| | Transiting | FN | TP |

where TN, FN, FP, TP stand for True Negative, False Negative, False Positive, True Positive, respectively.

The Confusion Matrix does not provide an explicit indicator for the performance of a classifier but is the basis to determine all other indicators used in the following analyses.

*Accuracy* Accuracy represents the percentage of correctly classified instances.

$$Accuracy = \frac{TP + TN}{\# \text{ of instances}}$$

Since the datasets are highly unbalanced, this value may represent a misleading KPI.

*Recall* Recall (or Sensitivity) is an indicator that specifies how often a patient that will actually pass to the SP phase is correctly classified. Formally it is defined as follows:

$$Recall = \frac{TP}{TP + FN}$$

This measure represents one of the most important KPIs since higher values correspond to a classifier able to correctly detect a wider portion of the transitioning patients.

*Precision* Precision (or Positive Predictive Value) represents the fraction of times the predictor classified a patient as transitioning to the SP phase and it was correct. It is defined as:

$$Precision = \frac{\text{TP}}{\text{TP} + \text{FP}}$$

Basically, the indicator represents the probability of having the target condition given a positive prediction. This measure is also of interest since higher values of this KPI correspond to a classifier less prone to classifying as transitioning a patient who in reality is not.

*Specificity* Specificity indicates how often a non-transitioning patient is actually correctly classified. Formally it is defined as follows:

$$Specificity = \frac{\text{TN}}{\text{TN} + \text{FP}}$$

Specificity can be considered as a counterpart of the Recall. Higher values of this metric correspond to a classifier able to correctly detect a wider portion of the non-transitioning patients. This indicator may be biased towards high values by the unbalanced nature of the data.

## Results

### Visit-Oriented setting

The results obtained by the various classifiers at different times are reported in Table 4 and S1 Table. Overall, Random Forests, SVM and AdaBoost performed interestingly well by leading to Recall values always higher than 75% with peaks of 100%, which means that none of the SP transitioning patient was not detected, at the expense of having small values of Precision, which are never higher than 10%. On the other hand, KNN seems not to be properly able to generalize its knowledge to new patients. Among the different algorithms, RF performed particularly well with an average recall value of 88% and a maximum value of 100%.

Considering Specificity and Accuracy, these two metrics are very similar, due to the large number of non-transitioning patients. Overall, all the models reach high values of both Accuracy and Specificity meaning that all the classifiers are able to correctly detect a wide portion of the non-transitioning patients. It is interesting to compare the results obtained in the two different kinds of datasets (FS and RS), namely when different types of information and amount of data are considered. First of all, the Precision values obtained with the RS dataset are higher than those obtained with the FS dataset, meaning that having a high number of visits is more important than having more features in order to reduce the number of FP (patients incorrectly classified as passing to the SP phase). If instead, the Recall values are considered, it turns out that the information brought by Status T1, Status T2 and Oligoclonal Banding are essential to obtain very high values of Recall for predictions within 180 days, while they are not so influential for more long-term predictions.

Furthermore analysing the predictions performance when varying the time-window, it is evident that short-term predictions are more reliable than long term predictions with better values of both Recall and Precision. This is a rational result considering that for long-term predictions the amount of uncertainty is higher. The only exception is represented by RS 720 where the overall performance of all models is unexpectedly higher than for shorter-term predictions.

### History oriented setting

The results obtained by the LSTM model for all the 6 datasets considered are reported in Table 5 and in S2 Table. Overall, LSTM yields Precision values higher than 42.7%, at the

**Table 4. Results of Visit-Oriented models on Feature-saving and Record-saving datasets at different time points.**

| Model | Feature-saving | | | | Record-saving | | | |
|---|---|---|---|---|---|---|---|---|
| | Accuracy | Recall | Precision | Specificity | Accuracy | Recall | Precision | Specificity |
| **180 days** | | | | | | | | |
| SVM | 86.4% | 94.4% | 5.5% | 86.4% | 87.2% | 85.0% | 8.6% | 87.2% |
| RF | 86.5% | 100.0% | 5.8% | 86.4% | 85.1% | 90.8% | 7.9% | 85.0% |
| AB | 86.1% | 100.0% | 5.6% | 86.0% | 85.4% | 88.9% | 7.9% | 85.3% |
| KNN | 72.6% | 80.6% | 2.4% | 72.6% | 85.6% | 81.2% | 7.4% | 85.7% |
| **360 days** | | | | | | | | |
| SVM | 85.1% | 86.5% | 4.9% | 85.1% | 86.6% | 80.7% | 8.2% | 86.7% |
| RF | 87.3% | 89.2% | 5.9% | 87.3% | 83.2% | 88.4% | 7.2% | 83.1% |
| AB | 85.5% | 83.8% | 4.9% | 85.5% | 83.6% | 88.4% | 7.3% | 83.5% |
| KNN | 71.2% | 67.6% | 2.0% | 71.2% | 85.0% | 77.3% | 7.1% | 85.1% |
| **720 days** | | | | | | | | |
| SVM | 84.8% | 81.1% | 4.8% | 84.8% | 87.8% | 77.3% | 9.3% | 87.9% |
| RF | 86.2% | 78.4% | 5.2% | 86.3% | 86.2% | 84.1% | 8.9% | 86.2% |
| AB | 86.9% | 70.3% | 4.9% | 87.1% | 85.0% | 84.5% | 8.3% | 85.1% |
| KNN | 75.1% | 64.9% | 2.4% | 75.2% | 85.2% | 75.8% | 7.6% | 85.4% |

SVM: Support vector machines; RF: Random Forest; AB: Ada Boost; KNN: K nearest neighbours

expense of values of the Recall metric, definitely lower than obtained with the VO approach. As in the visit-oriented approach, Precision values increase when considering the datasets with more visits, highlighting how, in order to improve the precision of these models, more data are needed. It is interesting to underline how, differently from the other models, LSTM performance improves when considering predictions over longer time-windows, suggesting that this approach may be more successful for more distant predictions, which are the most useful from a clinical point of view.

The low Recall values provided by LSTM models may be due to two different reasons. First of all, LSTMs are defined by a much more complex model if compared to the visit-oriented methods and so the learning phase requires a quantity of data which was not available in this study. Furthermore, LSTM predictions might have been harmed by the specific learning procedure used. Indeed, due to computational constraints, the LOGO procedure was not implemented but a train-test split was performed, reducing the number of instances available for the training process. It is tempting to speculate that this model might become more effective if used with many more and possibly less unbalanced data, in order to properly train these models and avoid the resource-consuming LOGO procedure.

**Table 5. Results of History-Oriented setting on Feature-saving and Record-saving datasets at 180, 360 and 720 days.**

| Days | Feature-saving | | | | Record-saving | | | |
|---|---|---|---|---|---|---|---|---|
| | Accuracy | Recall | Precision | Specificity | Accuracy | Recall | Precision | Specificity |
| 180 | 96.1% | 44.4% | 10.5% | 96.6% | 98.0% | 38.5% | 30.8% | 98.8% |
| 360 | 97.0% | 40.0% | 14.8% | 97.6% | 97.5% | 50.0% | 29.5% | 98.2% |
| 720 | 97.1% | 60.0% | 20.7% | 97.5% | 98.0% | 67.3% | 42.7% | 98.5% |

Results of Long short term memory model

## Threshold analysis

We next examined how the performance of the various models changes when only predictions with high certainty are considered. Indeed, for each prediction, the classifier returns a probability between 0 and 1. If the returned probability is over a certain threshold $\theta$ (set by default to 0.5), the classifier assigns the patient to the transitioning class ($\hat{y} = 1$), otherwise it assigns the patient to the non-transitioning class ($\hat{y} = 0$). The probability value returned by the model represents the level of "confidence" of the classifier in assigning a certain patient to a specific class: values closer to 0 and 1 correspond to more certain predictions returned by the model, while values closer to 0.5 represent those cases where the model is more uncertain on the output. Since a higher confidence of the models might correspond to better performances, focusing only on those predictions on which the models are more confident might improve the prognostic performance. In order to do that, we introduced a *confidence threshold* defined as:

$$CT_i = |p_i - 0.5|$$

which represents the level of confidence with which the model predicts the belonging class of a patient. In other words, given a record $i$, the model returns a probability $p_i$ for the patient to pass to the SP phase, and the confidence threshold is given by the distance between the predicted probability for that sample $p_i$ and the uncertainty value 0.5. Therefore, $CT$ will vary between 0 and 0.5, where 0.5 corresponds to a model 100% sure of its prediction and 0 corresponds to a model completely uncertain about the class of that sample. To make it clearer, suppose a model is highly confident that a patient will pass to the SP phase and returns a probability of $p_j = 0.9$. The corresponding confidence threshold will be $CT_j = 0.4$. On the other hand, if the probability returned by the model is of 0.51, $CT = 0.01$, indicating that the model is uncertain about that prediction.

In order to study how confidence on predictions influences the performance of the models, we focused on the the history-oriented approach (LSTM) and the most performing method for the visit-oriented approach (RF), using the Record-Saving datasets. We studied the performance trajectory of the models for different values of the confidence threshold. To be more specific, fixed a certain $CT$, we only considered the predictions with a distance from the uncertainty value (0.5) greater or equal to the confidence threshold. All other predictions were discarded. This essentially implies considering only those visits on which the models were sufficiently confident on the prediction. The performance metrics were then evaluated as described in the previous sections. This procedure was iterated many times, varying the confidence threshold between 0 and 0.49 and analyzing the performance of the models at each $CT$ value. Note that for $CT = 0$, the results are those reported in the previous sections.

Fig 2 represents the main KPIs and the percentage of patients belonging to class 0 and 1 for different confidence thresholds. Accuracy values are not plotted since they are very similar to Specificity values and, as already said, do not provide relevant insight into models performance.

First, the graphs keep track of the fraction of positive and negative samples left in the datasets while varying the confidence level. In RF models the percentage of samples related to transitioning and non-transitioning patients drops at similar rate when increasing $CT$. Conversely, for LSTM, the percentage of transitioning patients declines very rapidly as compared with non-transitioning patients. In other words, this model is able to correctly classify at least 80% of non-transitioning patients even with $CT > 0.45\%$. It is important to consider that the number of entries (records or time series) related to transitioning patients declines as confidence threshold increases. Thus, metrics are evaluated on a decreasing number of predictions, likely explaining the erratic behaviours of Precision and Recall seen at threshold values approaching

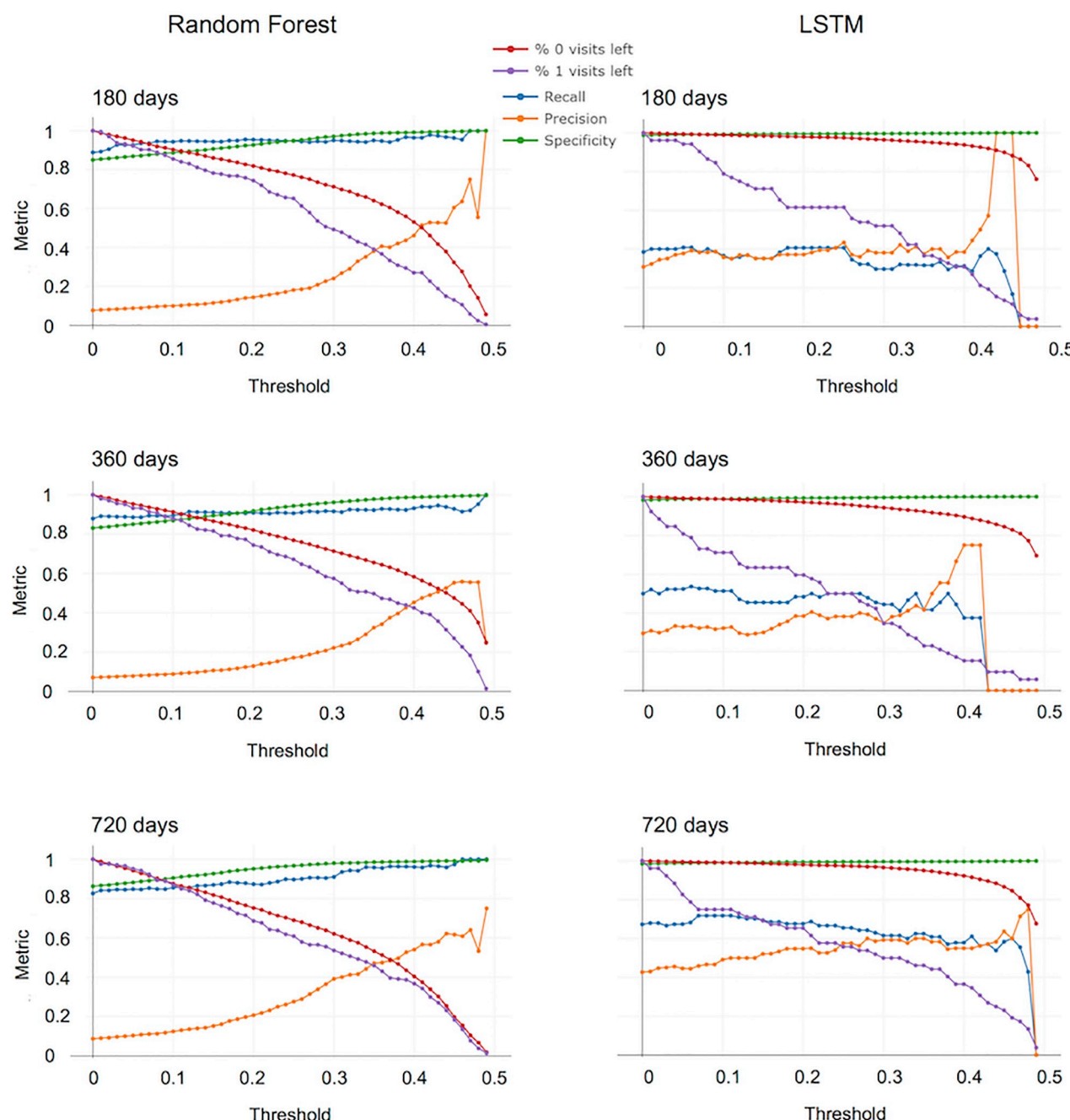

**Fig 2. Threshold analysis.** The choice of confidence threshold has different effects on the indicated variables. Analysis was performed with Random Forest and LSTM for Record Preserving datasets.

0.5. In particular, Recall and Precision are 0 when the number of TP is 0 and the metric is ill-defined. Recall values of RF model are always greater than 0.8, and increase for high confidence thresholds. This behaviour is observed at all time points. For LSTM, on the other hand, Recall values improve when dealing with longer time-windows, independently of the threshold chosen. At all time points, Recall values are little affected by *CT*, although a small increase is

observed at 720 days when the most casual predictions (corresponding to threshold below 0.05) are removed.

Concerning the Precision, the performance of both classifiers improves for higher levels of confidence. For low thresholds, Precision values are higher for LSTM than for RF, but values become quite similar when confidence threshold increases. It implies that for high confidence thresholds, the percentage of patients incorrectly classified as transitioning (FPs) decreases. Notably, at 720 days, about 50% of the total number of entries is classified with a confidence threshold of about 0.35 obtaining Precision values above 0.5.

Finally, both models yield high Specificity values for all the threshold levels, touching values of 100% for very high thresholds. This means that the two models are able to correctly detect the non-transitioning patients. However, as already said, this result may be due to the fact that data are unbalanced and the number of non-transitioning patients is far bigger than the number of transitioning.

Overall, the graphs suggest that RF is a stable and well performing model. Its ability to detect individuals going to transition to the SP phase (TP) is always above 0.80 and further increases when predictions with low confidence values are neglected. Increasing the confidence threshold, the number of records incorrectly classified as pertaining to transitioning patients (FP) is reduced. This partially overcomes the problem of very low Precision values yielded by all models used for the VO approach. By contrast, the performance of LSTM is little affected by the values of the confidence threshold. As already mentioned, this behaviour may be due to the lack of enough data for a proper training phase of these very complex models.

This last analysis highlights how RFs and LSTMs have different properties and are able to catch different aspects of the phenomenon under consideration, hence an interesting approach could be the possibility of combining the predictions returned by the two models in order to end up with a unique algorithm able to leverage the strengths of each model in a stacking-fashion [51].

## Discussion

In this paper we explored different machine learning approaches to obtain predictions on MS course in single patients, uniquely using real world data, normally available in clinical practice. Differently from previous works [22, 25, 27] we did not consider in this study imaging data specifically collected or analyzed for experimental purposes, and we present extensive numerical results using different ML forecasting models, considering various prediction settings and time horizons.

The definition of the two datasets (FS and RS) allowed to understand the role of some features on the models performance and the importance of the amount of available data for training the ML models.

The good Recall values obtained at 180 days in the VO setting using the FS dataset, which contains the results of liquor and MRI exams routinely rated by neuroradiologists show that considering imaging data makes visit-oriented models capable of unequivocal identification of patients going to transition to the SP phase within a short delay, as shown by the high/low percentage of TP/FN forecasts. However, for predictions farther away in time, the use of RS datasets (which do not include liquor and MRI results) improves Recall, suggesting that overall clinical conditions are good indicators of a patient's course. The main weak point of the VO setting is the occurrence of many FP predictions, reflected into low Precision values. Although this limits the clinical value of this approach, we show that performance can be strongly ameliorated by threshold analysis.

Furthermore, from a technical point of view, the occurrence of FP predictions indicates that training procedures were adequate to overcome the unbalanced nature of the datasets: machines did not use "shortcuts" by predicting only "non-transitioning".

Looking at patients' history makes long-term predictions better than short-term ones, strongly arguing in favour of this type of approach to obtain clinically relevant forecasts for a disease that evolves over many years [52]. In our hands, the HO setting performed worse than the visit-oriented in terms of Recall (that is, the ability to identify transitioning patients), but better in terms of Precision, meaning that this method yields far less FP predictions.

It must be noted that in the history-oriented setting we had a very small sample of transitioning patients, and the LOGO training procedure was computationally too demanding. Possibly, the combination of these two factors has harmed the performance of LSTM model, but this approach might be more fruitfully used with different endpoints, that give rise to less unbalanced databases.

## Conclusion

In summary, should our models analyze the clinical data of a new patient in RR phase and predict no transition to SP phase within two years from now, they would be right (as shown by Specificity indicator) in about 85% of the cases overall, and in almost 100% of the cases with a confidence threshold above 0.35. If used for support to therapeutic decisions, this means that there would be a very limited risk to under-treat patients at risk of rapid disease evolution. The situation is more shadowed for predictions of transition to the SP phase: although almost all the really transitioning cases would be correctly classified (Recall indicator), at least 50% of the cases would be FP (Precision indicator), and hence there would be a significant proportion of over-treated patients.

It must be noted that even the longest time window used in this work (2 years) might be too short to be really useful in the clinical practice, considering the slow effect of some therapies. However, we also show that the history-oriented setting is extremely promising for long-term predictions, provided that suitable data are available for training of LSTM model. Thus, our results provide clear indications that large and well maintained clinical databases can profitably be used to predict the course of MS in individual patients, inciting physicians to devote some effort to archive their data in a format compatible with computer analysis. This is a widely hoped-for goal [30, 31] that preludes to more effective use of the models proposed in this paper.

## Supporting information

**S1 Table. Confusion matrices for the Visit-Oriented setting.**
(PDF)

**S2 Table. Confusion matrices for the History-Oriented setting.**
(PDF)

## Acknowledgments

We thank Elisa Gioia, who offered the cue for starting the project.

## Author Contributions

**Conceptualization:** Ruggiero Seccia, Silvia Romano, Marco Salvetti, Andrea Tacchella, Andrea Zaccaria, Andrea Crisanti, Francesca Grassi, Laura Palagi.

**Data curation:** Silvia Romano, Anna Chiara Landi, Marco Salvetti, Francesca Grassi.

**Formal analysis:** Ruggiero Seccia, Daniele Gammelli.

**Software:** Ruggiero Seccia, Daniele Gammelli, Fabio Dominici.

**Supervision:** Francesca Grassi, Laura Palagi.

**Visualization:** Ruggiero Seccia, Fabio Dominici.

**Writing – original draft:** Ruggiero Seccia, Daniele Gammelli, Francesca Grassi, Laura Palagi.

**Writing – review & editing:** Ruggiero Seccia, Francesca Grassi, Laura Palagi.

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
