## [Decision Letter · Decision Letter 0]

7 Jan 2020

PONE-D-19-27834

Considering patient clinical history impacts performance of machine learning models in predicting course of multiple sclerosis

PLOS ONE

Dear Dr. Grassi,

Thank you for submitting your manuscript to PLOS ONE. After careful consideration, we feel that it has merit but does not fully meet PLOS ONE’s publication criteria as it currently stands. Therefore, we invite you to submit a revised version of the manuscript that addresses the points raised during the review process.

We would appreciate receiving your revised manuscript by Feb 21 2020 11:59PM. To enhance the reproducibility of your results, we recommend that if applicable you deposit your laboratory protocols in protocols.io, where a protocol can be assigned its own identifier (DOI) such that it can be cited independently in the future. For instructions see: http://journals.plos.org/plosone/s/submission-guidelines#loc-laboratory-protocols

We look forward to receiving your revised manuscript.

Kind regards,

Francesco Lolli, M.D., Ph.D.

Academic Editor

PLOS ONE

Journal Requirements:

2. In the Methods section and the online submission form, please provide additional information about the patient records used in your retrospective study. Specifically, please ensure that you have discussed whether all data were fully anonymized before you accessed them and/or whether the IRB or ethics committee waived the requirement for informed consent. If patients provided informed written consent to have data from their medical records used in research, please include this information.

4. Please ensure that you have provided details of the software used to implement the machine learning algorithms used in your study.

Reviewers' comments:

Reviewer's Responses to Questions

**Comments to the Author**

1. Is the manuscript technically sound, and do the data support the conclusions?

Reviewer #1: Yes

2. Has the statistical analysis been performed appropriately and rigorously? 

Reviewer #1: Yes

3. Have the authors made all data underlying the findings in their manuscript fully available?

Reviewer #1: Yes

4. Is the manuscript presented in an intelligible fashion and written in standard English?

Reviewer #1: Yes

5. Review Comments to the Author

Reviewer #1: 1. I would like the authors to be more explicit on the specific factors that predicted SPMS, and how that compares to the literature on predictors of SPMS. The authors should provide a brief review of that literature, to put this article into context.

2. Was there any subgroup analysis performed?

3. Are there plans for an independent validation set?

4. Two years is not much of a lag phase; is there any suggestion from records review that these patients had entered the SP stage before it was officially recognized?

5. What is the difference between 2nd line and immunosuppressant therapy? Please define these treatments (including 1st line).

6. What is the transition rate to SPMS so low (13% over 40 years) in this cohort?

6. PLOS authors have the option to publish the peer review history of their article (what does this mean?). If published, this will include your full peer review and any attached files.

Reviewer #1: No

---

## [Author Response · Author response to Decision Letter 0]

10 Feb 2020

RESPONSES TO JOURNAL REQUIREMENTS:

The manuscript has been prepared using the provided templates.

2. In the Methods section and the online submission form, please provide additional information about the patient records used in your retrospective study. Specifically, please ensure that you have discussed whether all data were fully anonymized before you accessed them and/or whether the IRB or ethics committee waived the requirement for informed consent. If patients provided informed written consent to have data from their medical records used in research, please include this information.

Anonymization is described at lines 137-138; the written consent is mentioned at lines 115-116.

We have uploaded the datasets used to FigShare, link will be provided upon acceptance of the paper. Please note that we have removed the labels to most features, to protect privacy of study participants to the maximum possible extent. This action will not preclude duplication of our results.

4. Please ensure that you have provided details of the software used to implement the machine learning algorithms used in your study.

The software is indicated at lines 231 and 273-274.

RESPONSES TO REVIEWER 1

1. I would like the authors to be more explicit on the specific factors that predicted SPMS, and how that compares to the literature on predictors of SPMS. The authors should provide a brief review of that literature, to put this article into context.

Thank you for your suggestion. Several sentences and references have been added to comply with your request. In particular, we have modified the manuscript at lines: 7-31; 41-42; 188-191. The references quoted in these paragraphs have been added.

2. Was there any subgroup analysis performed?

Unfortunately, our sample was too small for reliable subgroup analysis.

3. Are there plans for an independent validation set?

Thank you for raising this important point. We are currently working to extend our analysis to larger cohorts. The possible future development of the present work are now indicated at lines 104-108.

4. Two years is not much of a lag phase; is there any suggestion from records review that these patients had entered the SP stage before it was officially recognized?

Of course, SP phase is diagnosed retrospectively, and some delay is possible. However, we believe that the database is reasonably accurate, as patients' records are updated at every visit, and the disease phase can be revised if the attending neurologist thinks it is adequate to do so. This is now indicated at lines 120-121.

5. What is the difference between 2nd line and immunosuppressant therapy? Please define these treatments (including 1st line).

The treatments are now defined at lines 161-170.

6. What is the transition rate to SPMS so low (13% over 40 years) in this cohort?

Thank you for your observation. This important point is discussed at lines 124-133.

---

## [Editor Report · Decision Letter 1]

25 Feb 2020

Considering patient clinical history impacts performance of machine learning models in predicting course of multiple sclerosis

PONE-D-19-27834R1

Dear Dr. Grassi,

We are pleased to inform you that your manuscript has been judged scientifically suitable for publication and will be formally accepted for publication once it complies with all outstanding technical requirements. I made the final review myself.

With kind regards,

Francesco Lolli, M.D., Ph.D.

Academic Editor

PLOS ONE

Additional Editor Comments:

I have reviewed the manuscript myself. The authors have responded to all the questions and points from the previous review. The changes included are fit. The article in the present form is acceptable for publication.

---

## [Editor Report · Acceptance letter]

4 Mar 2020

PONE-D-19-27834R1 

Considering patient clinical history impacts performance of machine learning models in predicting course of multiple sclerosis 

Dear Dr. Grassi:

I am pleased to inform you that your manuscript has been deemed suitable for publication in PLOS ONE. Congratulations! Your manuscript is now with our production department. 

With kind regards,

on behalf of

Dr. Francesco Lolli 

Academic Editor

PLOS ONE